# Modulation of Motor Cortex Plasticity by Repetitive Paired-Pulse TMS at Late I-Wave Intervals Is Influenced by Intracortical Excitability

**DOI:** 10.3390/brainsci11010121

**Published:** 2021-01-18

**Authors:** George M. Opie, Ryoki Sasaki, Brodie J. Hand, John G. Semmler

**Affiliations:** Discipline of Physiology, Adelaide Medical School, The University of Adelaide, Adelaide 5005, Australia; george.opie@adelaide.edu.au (G.M.O.); ryoki.sasaki@adelaide.edu.au (R.S.); brodie.hand@adelaide.edu.au (B.J.H.)

**Keywords:** ageing, corticospinal descending volley, transcranial magnetic stimulation, motor cortex, short-interval intracortical facilitation, I-wave periodicity repetitive TMS

## Abstract

The late indirect (I)-waves recruited by transcranial magnetic stimulation (TMS) over primary motor cortex (M1) can be modulated using I-wave periodicity repetitive TMS (iTMS). The purpose of this study was to determine if the response to iTMS is influenced by different interstimulus intervals (ISIs) targeting late I-waves, and whether these responses were associated with individual variations in intracortical excitability. Seventeen young (27.2 ± 6.4 years, 12 females) healthy adults received iTMS at late I-wave intervals (4.0, 4.5, and 5.0 ms) in three separate sessions. Changes due to each intervention were examined with motor evoked potential (MEP) amplitudes and short-interval intracortical facilitation (SICF) using both posterior-anterior (PA) and anterior-posterior (AP) TMS current directions. Changes in MEP amplitude and SICF were influenced by iTMS ISI, with the greatest facilitation for ISIs at 4 and 5 ms with PA TMS, and 4 ms with AP TMS. Maximum SICF at baseline (irrespective of ISI) was associated with increased iTMS response, but only for PA stimulation. These results suggest that modifying iTMS parameters targeting late I-waves can influence M1 plasticity. They also suggest that maximum SICF may be a means by which responders to iTMS targeting the late I-waves could be identified.

## 1. Introduction

Transcranial magnetic stimulation (TMS) is a non-invasive brain stimulation technique that has significantly facilitated our ability to investigate human neurophysiology in vivo. In particular, TMS remains the only technique able to both induce and measure neuroplastic changes within the brain [1,2]. Neuroplasticity refers to the brains ability to modify its intrinsic structural and functional connectivity and is a process that is heavily implicated in core neurological functions such as generating memories, learning new skills, and recovering from brain damage [3,4,5]. The application of TMS for understanding neuroplasticity is, therefore, critical for research in both healthy individuals and clinical populations alike. 

While it remains unclear exactly which neuronal elements are targeted by TMS, it is well established that stimulation over primary motor cortex (M1) results in the trans-synaptic activation of corticospinal neurons via recruitment of local interneuronal circuits [6]. The input of these networks can be visualised using invasive recordings from corticospinal neurons, and this approach has revealed that TMS to M1 produces a complex volley of activity involving several discernible waves [7,8]. The indirect generation of these responses has led to them being labelled as indirect (I)-waves: these are generally referred to as early (I1) and late (I2/3) in order of appearance, occur with a periodicity of ~1.5 ms, and reflect input from different excitatory intracortical networks [9,10]. In addition to invasive measures directly from the corticospinal neuron, I-wave characteristics can also be investigated non-invasively using a TMS paradigm referred to as short-interval intracortical facilitation (SICF) [11,12]. This involves pairs of stimuli that produce facilitation of the associated motor evoked potential (MEP) when the interstimulus interval (ISI) approximates the I-wave periodicity of 1.5 ms. The observed facilitation is thought to stem from the second stimulus of the pair (S2) causing discharge of I-wave generating interneurons that were depolarised (but not discharged) by the first stimulus (S1) [13]. In support of this, pharmacological TMS studies have reported that SICF is modulated by neurotransmitters thought to regulate I-wave activity, including gamma aminobutyric acid (GABA), dopamine, and noradrenaline (for review, see [14]).

Although there are currently a number of TMS-based interventions that are capable of inducing neuroplastic changes in M1 [2], only one was designed to specifically target the interneuronal motor networks responsible for I-wave generation. Referred to as I-wave periodicity repetitive TMS (iTMS), this paradigm applies a low frequency (i.e., 0.2 Hz) train of 180 paired-pulse stimuli over a 15 min period (for review, see [15]). Importantly, iTMS can modulate the excitability of the interneuronal networks generating different I-waves. For example, early I-wave (I1) excitability can be modified using a 1.5 ms ISI [16,17], whereas late I-wave (I3) excitability can be modified using a 4.5 ms ISI [18,19]. Using this approach, previous work has reported a robust facilitation of MEP amplitude [16,17,20] and an increase in the magnitude of SICF [16], which are used as markers of M1 plasticity. 

Whereas the prototypical I-wave periodicity of 1.5 ms is often applied within studies that utilise SICF or iTMS, there is some variability around this value among individuals [12,21,22]. Importantly, previous work suggests that this variance impacts the efficacy of iTMS targeting the early I-wave [23]. Despite this, the influence of iTMS interval when targeting late I-waves has not been investigated. This is a particularly important limitation as these circuits are known to be more temporally variable than early I-wave circuits [21,24], potentially increasing the impact of this confound on the response to iTMS targeting late I-waves. Furthermore, late I-waves have been more heavily implicated in neuroplasticity of M1 [21,24,25] and motor learning [26], suggesting greater repercussions of response variability for the development of interventions aiming to modulate these waves in a functionally meaningful way.

The aim of the current study was, therefore, to investigate if the response to iTMS is influenced by different ISIs targeting late I-waves, and whether these responses are associated with individual variations in excitability of these cortical interneuronal circuits. To achieve this, iTMS was applied at three different ISIs in separate sessions, and the effects on corticospinal excitability were related to baseline excitability of late I-waves, assessed using SICF at the same ISIs. Furthermore, growing evidence demonstrates that variations in the direction of current induced within the brain by TMS recruits different late I-wave populations that have unique functional relevance (for review, see [27,28]). In an attempt to more comprehensively assess these different intracortical circuits, SICF was recorded using both posterior-to-anterior (PA) and anterior-to-posterior (AP) induced currents. Given the previous findings demonstrating a greater response when using an ISI optimised to suit individualised early I-wave timing [23], we hypothesised that the response to iTMS would be influenced by ISI and would be associated with late I-wave excitability assessed with SICF.

## 2. Materials and Methods

Seventeen young (mean age ± standard deviation: 27.2 ± 6.4 years, 12 females) adults were recruited from the university and wider community to participate in the current study. Exclusion criteria included a history of neurological or psychiatric disease, or current use of psychoactive medication (sedatives, antipsychotics, antidepressants, etc.). All subjects reported being right-handed. All experimentation was approved by the University of Adelaide Human Research Ethics Committee and conducted in accordance with the declaration of Helsinki. Each subject provided written, informed consent prior to participation.

### 2.1. Experimental Arrangement

The protocol included 3 experimental sessions that were held at least 7 days apart (mean time between sessions ± standard deviation, 14.1 ± 9.3 days)(Figure 1). For the duration of each session, participants were seated in a comfortable chair with their right arm and hand supported. Surface electromyography (EMG) was recorded from the first dorsal interosseous (FDI) muscle of the right hand using two Ag-AgCl electrodes in a belly-tendon montage. An electrode attached over the ulnar styloid process grounded the electrodes. EMG signals were amplified (300×) and band-pass filtered (20 Hz high pass, 1 kHz low pass) using a CED1902 signal conditioner (Cambridge Electronic Design, Cambridge, UK), and digitized at 2 kHz using a CED1401 interface (Cambridge Electronic Design). Signal noise within the 50 Hz frequency band (associated with mains power) was also removed using a Humbug mains noise eliminator (Quest Scientific, North Vancouver, Canada). To facilitate muscle relaxation when required, real-time EMG signals were displayed under high gain (50 μV/division) on an oscilloscope placed in front of the subject.

### 2.2. Experimental Procedures

#### 2.2.1. Transcranial Magnetic Stimulation (TMS)

TMS was applied to the hand area of the left M1 using a figure-of-eight branding iron coil connected to two monophasic Magstim 200^2^ magnetic stimulators via a Bistim unit (Magstim, Dyfed, UK). All stimulation was applied over the location producing an optimum response in the FDI muscle (assessed with a PA induced current), with the coil held tangential to the scalp and at an angle of approximately 45° to the sagittal plane. This location was marked on the scalp for reference and continually checked throughout the experiment. Stimulation inducing an AP current was applied over the same cortical location, but with the coil rotated 180° relative to the PA orientation. For each session, all pre- and post-iTMS measures were recorded using both PA and AP induced currents. Following identification of the cortical target for stimulation, resting motor threshold (RMT) was defined as the minimum stimulus intensity producing an MEP response ≥50 µV in at least 5 out of 10 consecutive trials during complete relaxation of the right FDI [29]. The stimulus intensity producing an MEP of approximately 1 mV in amplitude when averaged over 20 trials (MEP_1mV_) was then identified.

#### 2.2.2. Short-Interval Intracortical Facilitation (SICF)

SICF was investigated in the resting FDI muscle using an established paired-pulse TMS protocol, which produces peaks of MEP facilitation that are compatible with I-wave latencies recorded from the epidural space [12]. Measures of SICF used an S1 set at the MEP_1mV_ intensity and S2 at 90% of RMT (both intensities set individually for the relevant coil orientation). To encapsulate the range of intervals prototypically associated with the third peak of facilitation, ISIs of 4.0, 4.5, and 5.0 ms were applied [30]. A total of 15 trials were recorded for each condition in a pseudo randomised order, and therefore a single SICF recording block consisted of 60 trials as follows: 15 single pulses set at MEP_1mV_ intensity and 45 SICF trials (15 for each of the 3 SICF ISIs). In each session, SICF was recorded using both PA and AP induced current directions in separate blocks at baseline, and these measures were then repeated 5 and 30 min after the iTMS intervention (Figure 1).

#### 2.2.3. I-Wave Periodicity Repetitive TMS (iTMS)

In accordance with previous literature [16,31], iTMS involved 180 pairs of stimuli applied in a PA coil orientation every 5 s, resulting in a total intervention time of 15 min. The intensity of stimulation was the same as that used to test SICF (S1 = MEP_1mV_ intensity, S2 = 90% RMT) and iTMS ISIs targeting the third SICF peak (i.e., 4.0, 4.5, and 5.0 ms) were applied in separate sessions. The order in which these different iTMS interventions were applied was randomised among participants. In order to mitigate the effects of coil heating during the intervention, ice packs were used to cool the coil prior to and during iTMS application. This ensured that the same coil could be used for all TMS measures.

### 2.3. Data Analysis

Analysis of EMG data was completed manually via visual inspection of offline recordings. For all recordings, traces showing muscle activity >30 μV peak-to-peak amplitude in the 100 ms prior to TMS application were excluded from the analysis. MEP amplitudes were measured peak-to-peak and expressed in mV. For SICF measures within each subject, individual conditioned MEP trials within each ISI were expressed as a percentage of the mean test alone MEP amplitude recorded at baseline [16]. Then, post-intervention single and paired-pulse values were expressed as a percentage of baseline data. MEP amplitudes recorded during each iTMS block were averaged over 10 consecutive stimuli, resulting in 18 blocks. As between session comparisons of response amplitude during the first iTMS block identified a significant difference between sessions that could confound interpretation of the response to iTMS (see below), MEP data from the intervention was consequently expressed as a percentage of the first 20 trials recorded during iTMS (i.e., the first 2 blocks), and collapsed over quartiles.

### 2.4. Statistical Analysis

Data normality was assessed using Kolmogorov–Smirnov tests. One-factor linear mixed model analysis with repeated measures (LMMRM) was used to compare baseline RMT, MEP1mV amplitude, and MEP_1mV_ intensity between iTMS sessions (iTMS_4.0_, iTMS_4.5_, and iTMS_5.0_). Two-factor LMM*_RM_* was used to compare baseline measures of SICF between sessions and ISI (SICF_4.0_, SICF_4.5_, and SICF_5.0_). One-factor LMM*_RM_* was used to compare raw MEP amplitude in the first iTMS block between sessions, whereas two-factor LMM*_RM_* was used to compare raw MEP amplitude during iTMS between sessions and blocks (B1–18). Two-factor LMM*_RM_* was also used to compare normalised MEP amplitude during iTMS between sessions and quartiles (1–4). For changes in MEP_1mV_ post intervention, two-factor LMM*_RM_* was used to compare values between sessions and time point (Post 5 and Post 30). Three-factor LMM*_RM_* was used to compare the change in SICF between iTMS sessions, ISI, and post-intervention time point. Data recorded with different coil orientations were assessed in separate models using the same factors. For all models, subject was included as a random effect, and significant main effects and interactions were further investigated with Bonferroni corrected custom contrasts. Linear regression analysis was used to assess the relationship between the magnitude of SICF recorded within each ISI at baseline and the response to iTMS measured within the same session. Linear regression was also used to assess the relationship between the maximum facilitation observed at baseline (identified numerically by visual inspection of the data) and the response to iTMS within the same session. Unless otherwise stated, data are presented as mean ± SEM. Significance was set at *p* < 0.05.

## 3. Results

All participants completed each experimental session without adverse event. However, four individuals demonstrated high thresholds for AP stimulation that meant it was not possible to achieve the required MEP_1mV_ with this current direction. Consequently, while data from all 17 participants are included for measures using a PA TMS current, measures using an AP TMS current includes data from 13 participants. Table 1 shows RMT, MEP_1mV_ intensity, and MEP_1mV_ amplitude measured at baseline using each coil orientation. No differences were found between sessions for any of these variables (all *p*-values > 0.05). Table 2 shows baseline SICF values recorded for each ISI and current direction in each iTMS session. For PA stimulation, although facilitation differed between ISIs (*p* = 0.04), no significant post hoc differences were found. Furthermore, there was no difference between iTMS sessions (*p* > 0.05) and no interaction between iTMS sessions and SICF ISI (*p* = 0.8). For AP stimulation, facilitation again differed between SICF ISIs (*p* = 0.02), with post hoc comparisons showing more facilitation with SICF_4.0_ than SICF_5.0_ (*p* = 0.03). However, there was no difference between iTMS sessions (*p* = 0.2) or interaction between iTMS sessions and SICF ISI (*p* = 0.9).

### 3.1. Changes in Motor Evoked Potential (MEP) Amplitude during iTMS

Figure 2A shows raw MEP amplitude during the intervention as compared with iTMS sessions using different ISIs. These responses varied between iTMS sessions (*p* < 0.0001), with post hoc comparisons showing the response following iTMS_5.0_ was reduced relative to both iTMS_4.0_ (*p* < 0.0001) and iTMS_4.5_ (*p* < 0.0001), but no difference between iTMS_4.0_ and iTMS_4.5_ (*p* = 0.2). the MEP amplitude also differed between blocks (*p* < 0.0001), with post hoc comparisons showing that responses in Block 9 were elevated relative to Block 1 (*p* = 0.01) and remained elevated until the end of the intervention (all *p*-values < 0.0002). However, there was no interaction between factors (*p* = 0.1). 

As MEP amplitude during the first iTMS block varied between sessions (*p* = 0.03), data were normalised to the average amplitude of the first 20 trials recorded during the intervention (i.e., block 1 + 2, see data analysis), and then compared between quartiles (Figure 2B). These data varied between iTMS sessions (*p* < 0.0001) and quartiles (*p* < 0.0001) and there was an interaction between factors (*p* = 0.001). While no differences were found between iTMS sessions for the first and second quartiles, the response to iTMS_4.5_ was reduced relative to iTMS_4.0_ during the third quartile (*p* = 0.02) and reduced relative to both iTMS_4.0_ (*p* < 0.0001) and iTMS_5.0_ (*p* < 0.0001) during the fourth quartile. For iTMS_4.0_, responses during the third and fourth quartiles were greater than both the first and second quartiles (all *p*-values <0.001). For iTMS_4.5_, responses during the third quartile were greater than the first quartile (*p* < 0.05). For iTMS_5.0_, responses within the fourth quartile were greater than during the first three quartiles (all *p*-values <0.02). 

### 3.2. Effects of iTMS on PA-Sensitive Circuits

Post-iTMS changes in normalised MEP amplitude and SICF recorded with a PA current direction are shown in Figure 3. For MEPs, the response to iTMS varied between sessions (*p* = 0.04), with post hoc comparisons showing that the change in MEP amplitude following iTMS_5.0_ was greater than following iTMS_4.5_ (*p* = 0.03), but not different between iTMS_4.0_ and iTMS_5.0_ (*p* = 0.7, Figure 3A). Despite this, there was no difference between time points (*p* = 0.9) or interaction between factors (*p* = 0.6). For SICF, there was an interaction between iTMS session and SICF ISI on the change in facilitation (*p* = 0.03, Figure 3B), with post hoc comparisons showing that the increase in SICF_4.0_ was greater following iTMS_5.0_ than iTMS_4.5_ (*p* = 0.008). However, no other main effects or interactions were found.

### 3.3. Effects of iTMS on AP-Sensitive Circuits

Post-iTMS changes in normalised MEP amplitude and SICF recorded with an AP current direction are shown in Figure 4. For MEPs, the effect of iTMS again varied between sessions (*p* < 0.0001), with post hoc comparisons showing that the response to iTMS_4.0_ was greater than both iTMS_4.5_ (*p* = 0.001) and iTMS_5.0_ (*p* < 0.0001), but no difference between iTMS_4.5_ and iTMS_5.0_ (*p* > 0.9, Figure 4A). However, responses did not differ between time points (*p* = 0.2) and there was no interaction between factors (*p* = 0.3). For SICF, changes in facilitation differed between iTMS sessions (*p* < 0.0001) and SICF ISI (*p* = 0.03), and there was an interaction between these factors (*p* = 0.01, Figure 4B). Post hoc comparisons showed that changes in facilitation following iTMS_4.0_ were greater than changes following both iTMS_4.5_ and iTMS_5.0_ for all SICF ISIs (all *p*-values < 0.0001). Furthermore, the change following iTMS_4.0_ was greater for SICF_4.0_ than SICF_4.5_ (*p* < 0.001). An interaction between iTMS session and time was also found (*p* = 0.03, Figure 4C), with post hoc comparisons showing that the response to iTMS_4.0_ was greater than that of both iTMS_4.5_ and iTMS_5.0_ at both Post 5 and 30 time points (all *p*-values <0.0001). In addition, the response to iTMS_5.0_ was reduced at Post 5 time point relative to Post 30 time point (*p* = 0.02).

### 3.4. Linear Regression Analysis

The magnitude of baseline SICF within each ISI was not related to the response to iTMS for the same ISI for either TMS coil orientation. In contrast, irrespective of the ISI at which it occurred, the maximum SICF with PA TMS observed at baseline predicted post-intervention PA SICF following both iTMS_4.5_ (*r*^2^ = 0.4, *p* = 0.01, Figure 5A) and iTMS_5_ (*r*^2^ = 0.4, *p* = 0.01, Figure 5C), whereas maximum PA SICF measured at baseline also predicted the change in PA MEP amplitude following iTMS_5_ (*r*^2^ = 0.3, *p* = 0.04, Figure 5B).

## 4. Discussion

The current study assessed if the response to iTMS is influenced by different ISIs targeting the late I-wave circuits, and whether these responses are associated with individual variations in the excitability of these intracortical circuits. This was achieved by applying iTMS at three ISIs that characterise the prototypical timing of the third SICF peak and by assessing how the response to each condition related to baseline measures of SICF tested at the same ISIs. During the intervention, although all intervals produced an increase in MEP amplitude, the smallest response was seen following application of the 4.5 ms ISI that is most commonly associated with the peak of late I-wave excitability. Furthermore, while PA-sensitive circuits showed a non-specific increase in excitability following all iTMS intervals, AP-sensitive circuits showed a more specific increase that was restricted to the 4.0 ms ISI. Finally, the maximum SICF observed at baseline (irrespective of timing) predicted the response to iTMS, but this was only apparent for the response to PA stimulation.

### 4.1. Effects of iTMS on PA-Sensitive Circuits

For all ISIs, iTMS resulted in progressive increases in MEP amplitude during the intervention, in addition to a potentiation of PA MEPs following the intervention. Taken together, this suggests that irrespective of the specific ISI, the intervention was able to induce a neuroplastic modulation of excitability in PA-sensitive circuits in M1. While these changes are in keeping with the growing body of literature that has applied iTMS [15], the majority of previous studies have used ISIs targeting the early I-wave. Despite this, there has been some limited investigation of iTMS targeting the late I-wave (presumably I3) from our group [19] and others [18]. While findings of the current study are consistent with our previous study, which found a potentiation of corticospinal excitability following iTMS using ISIs of 4.1 and 4.9 ms, they are in contrast to the response reported by Long, Federico, Perez [18], which found that iTMS with a 4.3 ms ISI did not modulate PA-sensitive MEPs. While the intensity and number of stimuli applied during iTMS was consistent among all of these studies, the timing between stimulus pairs was notably different; whereas Long, Federico, Perez [18] applied 180 stimuli over ~30 min, both the current and previous studies from our group applied the same number of stimuli over 15 min. Consequently, it is possible that the frequency at which paired stimuli are applied is an important determinant of the response to iTMS. Furthermore, it has been previously suggested that a 30 min iTMS block may more strongly engage metaplastic regulatory mechanisms that could reduce the post-intervention potentiation of MEPs [16]. Therefore, differences in the duration and frequency of the intervention may have contributed to these contrasting findings. Despite the specific mechanism, these conflicting results demonstrate the need for further characterisation of how stimulation parameters influence the response to iTMS, as this space is relatively unexplored.

After allowing for differences in response amplitude at the start of the intervention, the smallest potentiation apparent during iTMS was seen for the 4.5 ms interval, with significantly larger responses apparent during iTMS_4.0_ and iTMS_5.0_ (Figure 2B). Furthermore, post-intervention changes in PA MEPs were also significantly reduced following iTMS_4.5_ (Figure 3A). As the 4.0 and 5.0 ms intervals correspond to the rising and falling flanks of the prototypical I3 peak, whereas the 4.5 ms interval corresponds to the expected point of optimum facilitation [30,32], this pattern of response is contrary to predicted outcomes. Given that baseline excitability of PA-sensitive circuits was numerically greatest for the 4.5 ms interval (see PA SICF_4.5_ in Table 2), one explanation for this outcome could be that further increases in the excitability of these circuits were limited by a ceiling effect. However, the difference in facilitation between intervals was not large, so it is possible that other factors that are not apparent here may have contributed. Nonetheless, these outcomes suggest that the response to iTMS targeting PA-sensitive late waves is temporally sensitive, at least at the group level. The functional relevance of this limited response remains to be determined.

Consistent with the observed changes in MEP amplitude, PA SICF also showed a moderate increase in facilitation post iTMS. However, this increase was generally consistent between iTMS ISIs, with only one small difference observed between the response of SICF_4.0_ to iTMS_4.5_ and iTMS_5.0_. As SICF is thought to more specifically index the excitability of the intracortical excitatory circuits responsible for I-wave generation, this may suggest that the differential potentiation of PA MEPs following each iTMS ISI stemmed from changes in other circuits. In particular, the 4–5 ms interval represents the tail of rapid intracortical inhibition mediated by GABA type-A receptors (GABA_A_) [14,33], suggesting that variable input from this circuit could confound the excitatory response to iTMS. Despite this, although the current study is the first to demonstrate a potentiation of PA SICF following iTMS targeting the I3-wave, a single previous study reported that iTMS targeting the I1-wave resulted in greater PA SICF across a wide range of ISIs that characterised all three peaks of facilitation [16]. As this study used an iTMS ISI approximating the synaptic delay period (i.e., 1.5 ms), the authors suggested that this broadband potentiation stemmed from an increase in the efficacy of synaptic connections between the interneuronal circuits responsible for generation of each I-wave volley and the corticospinal output neuron [16]. It could be suggested that the potentiation observed across SICF ISIs within the current study may reflect a similar effect. However, the narrow range of SICF ISIs applied here all fall within the period of the I3 peak, and therefore extrapolation beyond this is not possible.

### 4.2. Effects of iTMS on AP-Sensitive Circuits

In contrast to the generalised potentiation of PA MEPs, AP MEPs demonstrated a more specific increase that was limited to iTMS with a 4 ms ISI (iTMS_4.0_). While this response may at first appear to support a similar iTMS-induced potentiation of AP MEP amplitude at a 4.3 ms ISI that was reported by Long and colleagues [18], we found no MEP facilitation at a 4.5 ms interval, which complicates comparisons among studies. One factor that would influence how these studies can be compared is the temporal sensitivity of iTMS effects. While this parameter has not been specifically investigated, there is some evidence (albeit based on the I1-wave) that iTMS can produce corticospinal potentiation using an ISI that is an average of ~0.2 ms from the individual peak of SICF excitability [23]. As the excitability of AP-sensitive I3 circuits generally peaks around 4.0–4.1 ms (see Table 2 and [30,34]), facilitatory effects of iTMS ISIs ranging from ~3.8 to 4.3 could be reasonable for these circuits, whereas a facilitatory response to a 4.5 ms ISI would appear to be less likely. Consequently, it is possible that the findings from the current study and those of Long and colleagues provide complimentary information about the temporal resolution of iTMS effects targeting AP-sensitive circuits. 

Changes in AP SICF following iTMS were also very specific, with a large increase in facilitation again only apparent for the 4.0 ms ISI. One factor that may have contributed to this response is that the intensity used to apply iTMS was set relative to the PA current, which would have represented a lower proportion of the stimulation threshold required to activate AP-sensitive circuits [35,36]. Consequently, it is possible that the intervention was only able to modulate AP-sensitive circuits when applied at the peak of excitability within those circuits (i.e., ~4 ms, see above), resulting in a response that was restricted to iTMS_4.0_. Alternatively, the conventional interpretation of the response to AP stimulation is that it results in more specific recruitment of late I3-waves [35]. The specific effects of iTMS_4.0_ could, therefore, reflect a more temporally resolved neuromodulatory effect of iTMS. However, the literature developing around this topic in fact suggests that the intracortical circuits activated by different current directions are unique, particularly for the late I-waves (for review, see [27,28]). Consequently, it seems more likely that the specificity of the response reflects a characteristic of the activated interneuronal network. In support of this, recent work from our group that characterised late I3-wave excitability using SICF showed that AP stimulation produced a peak of facilitation that was more temporally constrained than the peak produced by PA stimulation [30]. 

Given the interpretation that responses produced by different current directions involve activation of unique intracortical circuits [27], it is interesting to note that although the intervention was only applied with a PA current, facilitation of responses generated by both PA and AP currents was observed. However, while varying current direction likely results in the recruitment of different interneuronal populations, the low spatial resolution and indiscriminate nature of the stimulus provided by TMS means that it is likely that there is overlap in the populations activated by different current directions. Consequently, it is reasonable to suggest that a modulatory intervention delivered in one current direction may still influence measurements made with different current directions, and this is supported by the findings reported here. Despite this, the extent of this overlap remains to be determined, and future research addressing this question will be an important component of developing functionally relevant neuromodulatory interventions based on iTMS. In particular, the waveform used to apply TMS can influence the neuronal networks activated by stimulation [34,37,38], which in turn would be expected to affect the neuromodulatory effects of iTMS. As stimulation within the current study was limited to a monophasic pulse with fixed characteristics (i.e., width and height), future studies investigating how different waveforms influence iTMS targeting the late I-waves will be important.

### 4.3. Baseline Facilitation Predicts iTMS Response, but Not in the Expected Way

Previous work has suggested that the neuromodulatory effects of TMS can be improved by modifying stimulation parameters to suit individual neurophysiological characteristics [23,39], and therefore we were interested to see if variations in baseline I-wave excitability influenced the response to iTMS. If the timing of I-wave excitability was an important determinant of the response to iTMS, we expected that the iTMS response would be greatest following an intervention that used an ISI that coincided with the individual timing of peak facilitation (i.e., the magnitude of baseline SICF within each ISI should predict the response to iTMS within the same ISI). What we actually found was that the iTMS response was predicted by the largest SICF value measured at baseline, irrespective of the timing at which this occurred (Figure 5A–C). Consequently, it appears that the timing of facilitation is less important for the response to iTMS than the absolute baseline excitability of these circuits. However, the extent of this relationship appeared to increase as the iTMS interval lengthened: while baseline measures failed to predict the response to iTMS_4.0_, greater maximum baseline SICF predicted a larger increase in maximum SICF following iTMS_4.5_ (Figure 5A), whereas greater maximum baseline SICF predicted greater potentiation of both MEPs (Figure 5B) and maximum SICF (Figure 5C) following iTMS_5.0_. Interestingly, examination of the timing at which maximum baseline SICF occurred within each iTMS session showed a coincidental skew of the distribution towards earlier intervals as iTMS ISI lengthened (Figure 5D). Consequently, while the underlying mechanism remains unclear, it seems that greater baseline late I-wave excitability at earlier intervals results in a better response to iTMS targeting the late I-waves, irrespective of the ISI at which iTMS is applied. One caveat to this outcome was that a relationship was only apparent for measures using a PA current, whereas all baseline AP measures were unrelated to effects of iTMS, (despite the facilitation observed with AP stimulation, Figure 4). As suggested above, it seems likely that there is overlap in the interneuronal populations activated by different current directions. Consequently, a potential explanation for this lack of relationship could be that the increased excitability observed with AP stimulation actually stemmed from a potentiation of PA-sensitive circuits that were also activated by AP stimulation; a possibility that will require confirmation in future research. Despite this, these outcomes suggest that measures of baseline SICF recorded with PA current could serve as a means of identifying individuals that are more likely to respond to iTMS targeting the late I-waves. This could be useful for optimising the application of neuromodulatory interventions in both healthy and clinical populations. 

## 5. Conclusions

In conclusion, while application of iTMS produced potentiation of the response to both PA and AP stimulation, the changes observed with each current direction demonstrated differential temporal specificity, with the greatest effects observed for ISIs at 4 and 5 ms with PA TMS, and 4 ms with AP TMS. These findings possibly reflect unique characteristic of different interneuronal populations. Furthermore, the response to iTMS targeting the late I-waves appeared to be greatest for individuals in whom baseline excitability of these circuits was increased at shorter intervals. This suggests that it may be possible to use baseline SICF characteristics to identify how well an individual might respond to iTMS targeting the late I-waves. However, it also suggests that optimising the intervention to target the individual timing of peak I-wave excitability is unlikely to produce an improved response.

## Figures and Tables

**Figure 1 brainsci-11-00121-f001:**
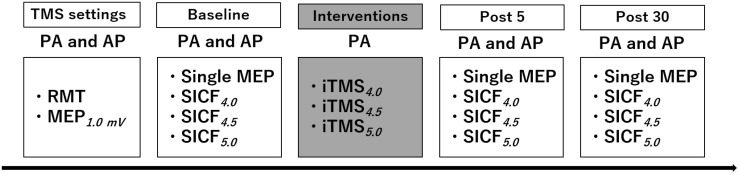
Experimental protocol.

**Figure 2 brainsci-11-00121-f002:**
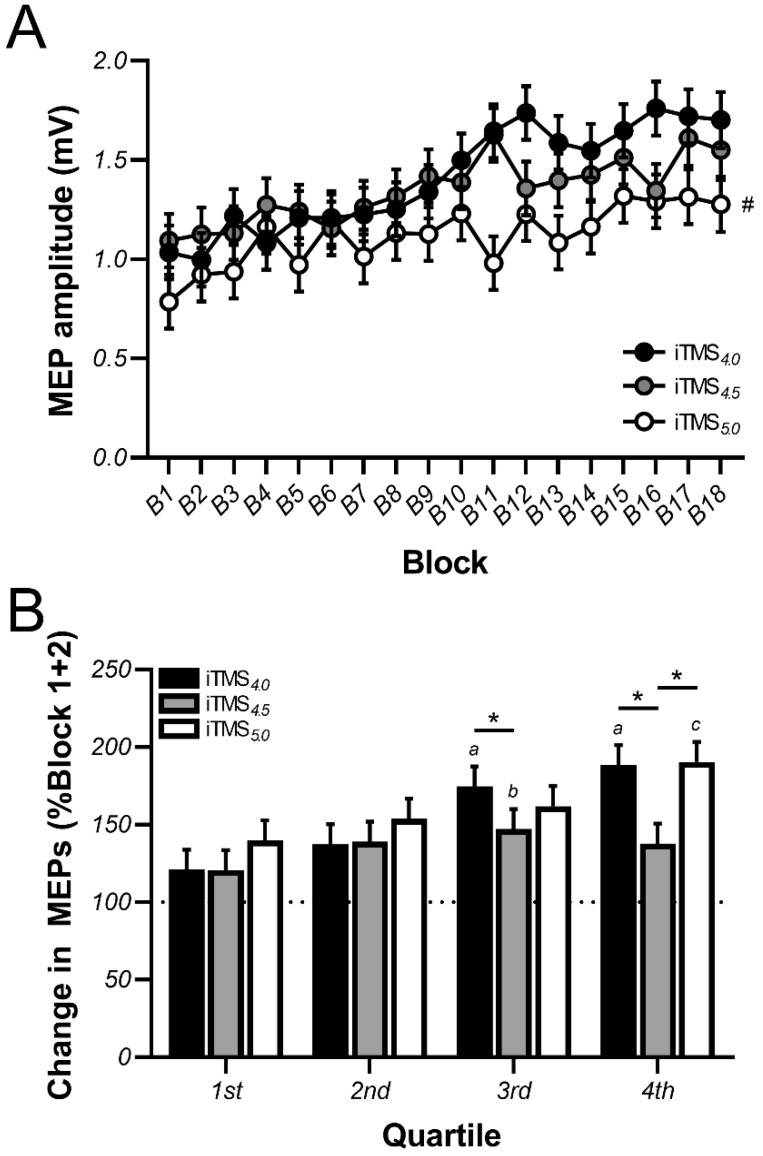
Corticospinal excitability is increased during iTMS. Changes in motor evoked potential (MEP) amplitude recorded during iTMS_4.0_ (black circles/bars), iTMS_4.5_ (grey circles/bars), and iTMS_5.0_ (white circles/bars) are shown as raw values averaged over 10 consecutive MEP trials, resulting in 18 blocks (**A**) or as normalised quartiles, expressed relative to the mean response recorded during the first 20 MEP trials (**B**). The dotted horizontal line in panel B represents no change in MEP amplitude, with an increase above this showing MEP facilitation. ^#^
*p* < 0.05 as compared with iTMS_4.0_ and iTMS_4.5_; ^a^
*p* < 0.05 as compared with the 1st and 2nd quartiles (for iTMS_4.0_ only); ^b^
*p* < 0.05 as compared with the 1st quartile (for iTMS_4.5_ only); ^c^
*p* < 0.05 as compared with the first 3 quartiles (for iTMS_5.0_ only); * *p* < 0.05.

**Figure 3 brainsci-11-00121-f003:**
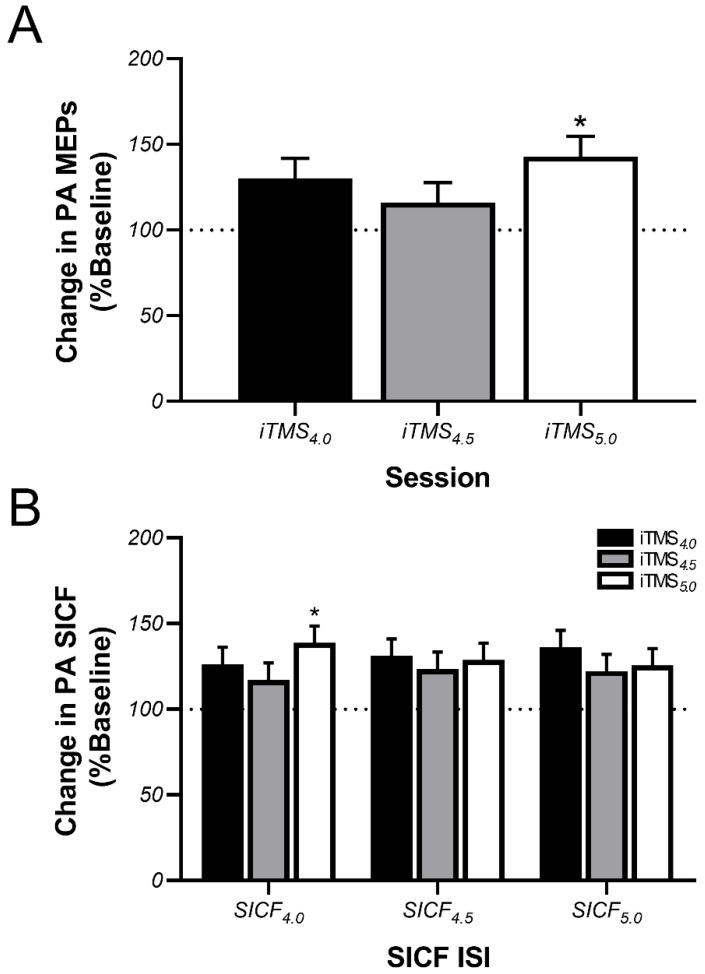
PA-sensitive circuits show a generalised increase in excitability following iTMS. Data show changes in MEP amplitude (**A**) and SICF (**B**) assessed with PA TMS following application of iTMS_4.0_ (black bars), iTMS_4.5_ (grey bars), and iTMS_5.0_ (white bars), expressed relative to baseline measures. The dotted horizontal line shows no change, with values above this line showing a potentiation of MEPs/SICF. * *p* < 0.05 as compared with iTMS_4.5_.

**Figure 4 brainsci-11-00121-f004:**
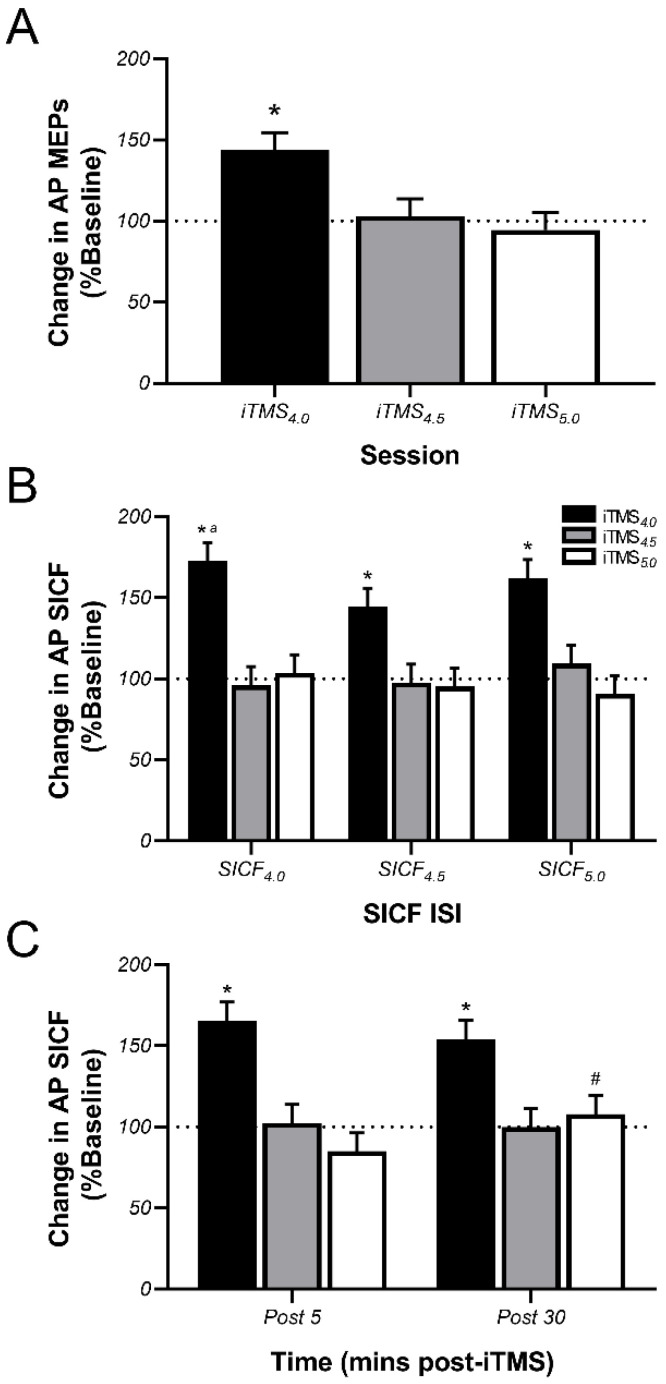
AP-sensitive circuits show a specific increase in excitability following iTMS. Data show changes in MEP amplitude (**A**) and SICF (**B**,**C**) assessed with AP TMS following application of iTMS_4.0_ (black bars), iTMS_4.5_ (grey bars), and iTMS_5.0_ (white bars), expressed relative to baseline measures. The dotted horizontal line shows no change, with values above this line showing a potentiation of MEPs/SICF. * *p* < 0.05 as compared with iTMS_4.5_ and iTMS_5.0_; ^a^
*p* < 0.05 as compared with SICF_4.5_ (for iTMS_4.0_ only); ^#^
*p* < 0.05 as compared with the Post 5 time point.

**Figure 5 brainsci-11-00121-f005:**
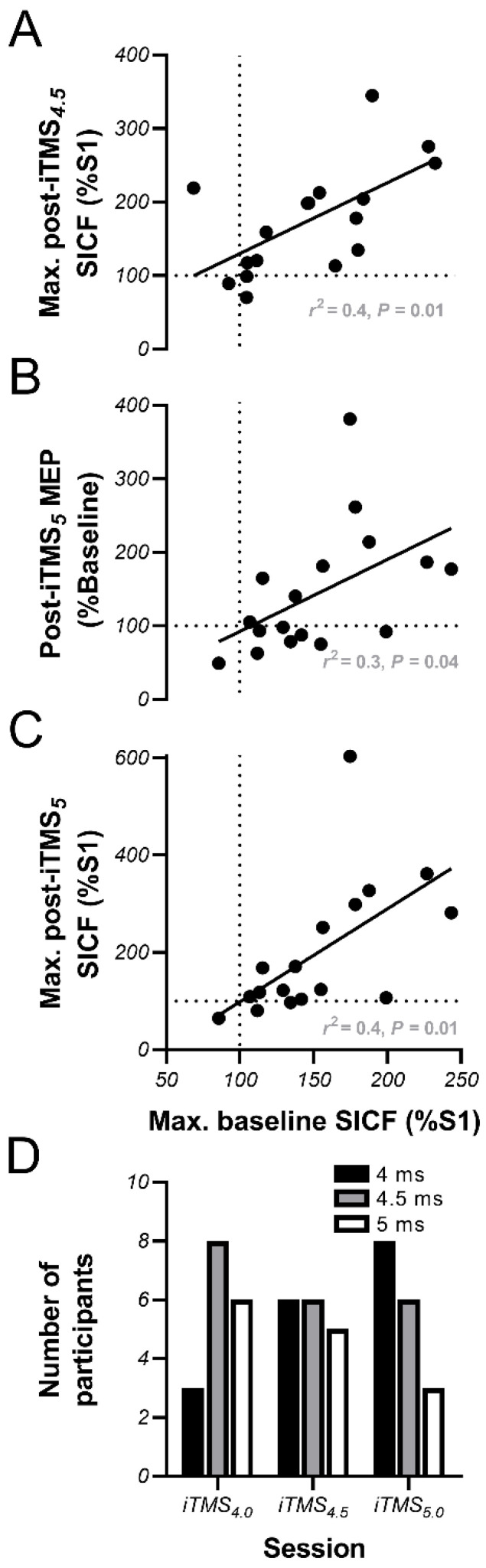
Maximum baseline SICF with PA TMS predicts the response to iTMS. Correlation data shows how the maximum SICF observed at baseline using PA TMS predicts changes in PA SICF (**A**,**C**) and MEP amplitude (**B**) following iTMS_4.5_ (**A**) and iTMS_5.0_ (**B**,**C**). Bar graph shows the distribution of SICF intervals at which individual participants demonstrated maximum facilitation at baseline within each iTMS session (**D**). The dotted lines in (**A**–**C**) show no change in response amplitude, with values above this line indicating response potentiation. As correlation analysis focused on the largest baseline SICF value out of the three different ISIs tested, the plotted points do not include all SICF values recorded.

**Table 1 brainsci-11-00121-t001:** Baseline single-pulse TMS characteristics for each I-wave periodicity repetitive transcranial magnetic stimulation (iTMS) session.

		iTMS_4.0_	iTMS_4.5_	iTMS_5.0_
RMT (% MSO)	PA	48.8 ± 1.9	48.1 ± 1.9	49.2 ± 1.9
	AP	64.8 ± 2.0 *	64.5 ± 2.0 *	65.2 ± 2.0 *
MEP_1mV_ intensity (% MSO)	PA	58.4 ± 2.7	58.5 ± 2.7	59.4 ± 2.7
	AP	77.7 ± 2.3 *	75.9 ± 2.4 *	77.5 ± 2.4 *
MEP_1mV_ amplitude (mV)	PA	0.8 ± 0.06	0.9 ± 0.06	0.8 ± 0.06
	AP	0.8 ± 0.06	0.7 ± 0.06	0.8 ± 0.06

* *p* < 0.05 compared to PA; MSO, maximum stimulator output.

**Table 2 brainsci-11-00121-t002:** Baseline short-interval intracortical facilitation (SICF) values for each iTMS session.

		iTMS_4.0_	iTMS_4.5_	iTMS_5.0_
PA	SICF_4.0_	111.1 ± 8.1	121.3 ± 8.2	123.4 ± 8.2
	SICF_4.5_	122.9 ± 8.2	136.6 ± 8.3	130.0 ± 8.2
	SICF_5.0_	111.7 ± 8.2	117. 5 ± 8.2	126.6 ± 8.2
AP	SICF_4.0_	122.2 ± 9.2	136.4 ± 9.2	117.4 ± 9.2
	SICF_4.5_	118.7 ± 9.2	130.0 ± 9.2	118.2 ± 9.1
	SICF_5.0_	99.0 ± 9.1	108.2 ± 9.1	109.3 ± 9.1

All values represent percent facilitation of the response to S1 + S2 relative to S1 alone.

## Data Availability

Data are available from the corresponding author upon request.

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
