# Peer review of "Modulation of Motor Cortex Plasticity by Repetitive Paired-Pulse TMS at Late I-Wave Intervals Is Influenced by Intracortical Excitability"

_brainsci, 2021, doi:10.3390/brainsci11010121_

Round 1

Reviewer 1 Report

This is an interesting and well written research about the interaction between intracortical facilitation and the modulation of M1 excitability by iTMS. I really appreciate the manuscript but some points should be clarified.

I suggest to introduce the reason to include both the P-A and A-P induced currents.

A short paragraph on potentially involved neurotransmitters would be desirable.

Finally, in the discussion, authors should refine their data in a more linear way introducing its usefulness in the design of rehabilitative protocols aimed at recovering lost synaptic plasticity.

Author Response

see attached document for response to comments

Reviewer 2 Report

In this manuscript, Opie et al. studied the neurophysiological effects of I-wave periodicity rTMS protocols with different interstimulus intervals (ISIs). Their principal findings were that different ISIs significantly affected both MEP amplitudes and SICF facilitation. The most extensive effects were received with ISIs of 4 and 5ms. The manuscript is well-written, summarizes the relevant previous literature, and the results support the conclusions. The findings are of interest to the readers of Brain Sciences and hold clinical potential. Below are my detailed comments.

  1. Table 1, abbreviation MSO has not been defined
  2. Figure 2 A. What is the difference between a block, a trial, and an epoch? Why are there only 18 MEPs but 10 trials?
  3. When the authors refer “aP < 0.05 compared to the 1st and 2nd quartiles”, does this mean that iTMS40 in the 3rd quartile differs from all iTMS protocols in 1st and 2nd quartiles or just other iTMS40 protocols? Same in other similar comparisons and other figures.
  4. How was the maximum SICF defined? For example, in Figure 5C, does this include all SICF results?
  5. To my understanding, the authors used monophasic waveform in their iTMS? The authors should also state that the applied waveform could influence the results. Also, the applied stimulation intensity could affect the effects.
  6. How many days apart were the experiments conducted?
  7. The authors should clarify the study design. I have read the sentence “A total of 15 trials were recorded for each condition in a pseudo randomised order, and a single SICF recording block therefore consisted of 48 trials (12 single pulses set at MEP1mV 131 intensity, 36 paired pulses).”, but I still do not understand how many pulses were given at each phase.
  8. To my understanding, the authors did the iTMS with a TMS device without cooling. If no cooling was used, it does not seem feasible that the same coil lasted the whole intervention. Was the coil changed within the protocol? If yes, have the authors tested that these coils are equally powerful? Sometimes coils induce different rMTs.
  9. The authors state that “MEP amplitude also differed between blocks (P < 0.0001), with post hoc comparisons showing that responses in block 9 were elevated relative to block 1 (P = 0.01), and remained elevated until the end of the intervention (all P-values < 0.0002).” Could the first MEP be higher due to the startle effect and had nothing to do with intervention response?
  10. The text seems to include some part of the Brain Sciences template on lines 258-260: “. This section may be divided by subheadings. It should provide a concise and precise description of the experimental results, their interpretation as well as the experimental conclusions that can be drawn.”
  11. I-wave latencies and effects are different on the group-level and the individual level. The authors might want to discuss this briefly.

Author Response

see attached document for responses
